# Evidencing New Roles for the Glycosyl-Transferase Cps1 in the Phytopathogenic Fungus *Botrytis cinerea*

**DOI:** 10.3390/jof8090899

**Published:** 2022-08-24

**Authors:** Matthieu Blandenet, Isabelle R. Gonçalves, Christine Rascle, Jean-William Dupuy, François-Xavier Gillet, Nathalie Poussereau, Mathias Choquer, Christophe Bruel

**Affiliations:** 1Microbiologie, Adaptation et Pathogénie, UMR5240, Univ Lyon, Université Lyon 1, Bayer SAS, 69622 Villeurbanne, France; 2Centre de Recherche La Dargoire, Bayer SAS, 69009 Lyon, France; 3Plateforme Protéome, Univeristy of Bordeaux, 33000 Bordeaux, France

**Keywords:** glycosyl-transferase, *Botrytis*, fungal cell wall, adhesion, secretomics

## Abstract

The fungal cell wall occupies a central place in the interaction between fungi and their environment. This study focuses on the role of the putative polysaccharide synthase Cps1 in the physiology, development and virulence of the grey mold-causing agent *Botrytis cinerea*. Deletion of the *Bccps1* gene does not affect the germination of the conidia (asexual spores) or the early mycelial development, but it perturbs hyphal expansion after 24 h, revealing a two-phase hyphal development that has not been reported so far. It causes a severe reduction of mycelial growth in a solid medium and modifies hyphal aggregation into pellets in liquid cultures. It strongly impairs plant penetration, plant colonization and the formation of sclerotia (survival structures). Loss of the BcCps1 protein associates with a decrease in glucans and glycoproteins in the fungus cell wall and the up-accumulation of 132 proteins in the mutant’s exoproteome, among which are fungal cell wall enzymes. This is accompanied by an increased fragility of the mutant mycelium, an increased sensitivity to some environmental stresses and a reduced adhesion to plant surface. Taken together, the results support a significant role of Cps1 in the cell wall biology of *B. cinerea*.

## 1. Introduction

Fungi are organisms that exhibit a saprophytic, symbiotic or parasitic lifestyle. In all cases, they undergo many biotic and abiotic stresses and interactions during their development. At the interface with their environment lies the fungal cell wall (FCW), a structure that plays a critical role in cell protection and in interaction with other organisms. The FCW is mainly (90%) composed of polysaccharides that are synthesized by membrane proteins of the glycosyltransferases (GTs) family [1,2]. Through cross-linking reactions by specific glycosylhydrolases, the polysaccharides form a matrix to which both GPI-anchored and secreted glycosylated proteins (O- or N-linked mannan or galactomannans) can attach to. These proteins carry important functions, such as signal transduction, adhesion, biofilm formation or cell wall remodeling [3].

FCWs have been described since the late 1970s, but thorough characterizations have only been performed in few species, such as *Aspergillus fumigatus*, *Saccharomyces cerevisiae*, *Candida albicans* or *Cryptococcus neoformans* [4]. In these model organisms, new GT activities (i.e., β-1,3-1,4-glucan synthase) and new structural descriptions continue to expand our knowledge of this important fungal structure [5,6], but the cell wall composition and function in other fungal species remain largely unknown.

The current model of cell wall polysaccharides biosynthesis in fungi proposes that it occurs at the plasma membrane, in the apical regions where the mycelium expand and in the distal regions where septa are built [7,8]. Among the 115 families of GTs inventoried in the carbohydrate active enzymes database (CAZy, http://www.cazy.org, August 2022 [9], 4 families group together enzymes able to produce polysaccharides: the GT2, GT5, GT48 and GH16 families. The polysaccharide synthases in these families are iterative enzymes that catalyze the transfer of sugar residues to a carbohydrate acceptor from an activated sugar-nucleotide donor [10]. By doing so they produce a nascent polysaccharide chain, and their supposed localization at the plasma membrane would allow the extrusion and extension of this chain into the FCW. The GT2 family contains the chitin synthases (CHS) that produce chitin, a hallmark polymer of the fungi that plays a structural role in the FCW [11]. The other fungal GT2s have received less attention so far.

In the plant pathogenic and necrotrophic models *Botrytis cinerea*, the FCW seems to undergo some modifications at the early step of the infection process, when hyphae develop into melanized infection cushions in which chitin appears to be modified into chitosan [12,13]. As in other pathogenic species, the cell wall of *B. cinerea* is expected to play an important role in virulence, and this is supported by (1) the discovery of inhibitors of CHS activity that protect plants against it [14] and (2) genetic studies that revealed a significant-to-essential role for CHS1, CHS3a, CHS6, and CHS7 in the development of *B. cinerea,* or in its interaction with plants [15,16,17,18,19,20]. Whether the other GT2-encoding genes would also play an important role in the biogenesis and/or remodeling of the FCW of this fungus has not been investigated.

Here, the role of one GT2-encoding gene in the biology of *B. cinerea* was studied. This gene is orthologous to the c*ps1/cpsA* genes characterized in the basidiomycete *C. neoformans* [21,22] and in the ascomycetes *Neurospora crassa*, *A. nidulans*, *Zymoseptoria tritici*, *Fusarium graminearum*, *Magnaporthe oryzae*, *A. fumigatus* and *F. verticilloides* [23,24,25,26,27,28,29,30]. In all of these fungi, the absence of *cps1* has a significant impact on mycelial development, cell wall content and resistance to stresses (Figure 1A). In the ascomycete fungi, the Cps1 protein is proposed to participate in cell wall biosynthesis, cell growth and host interaction, but its enzymatic function and its exact role have remained as elusive as its polysaccharide product. Our results in *B. cinerea* reveal new roles of Cps1 in FCW composition and integrity, hyphal growth, sclerotial development and mycelium adhesion to plant surface.

## 2. Materials and Methods

### 2.1. Strains and Culture Conditions

Fungal strains: *B. cinerea* B05.10 [teleomorph *Botryotinia fuckeliana* (de Bary) [Whetzel] and derived mutants. Bacterial strains: *Escherichia coli* One shot TOP10 (Thermofisher Scientific, C404010) and *Agrobacterium tumefaciens* LBA1126.

*E. coli* was grown at 37 °C in an LB medium (1% bactopeptone, 0.5% yeast extract, 0.5% NaCl) supplemented with kanamycin (50 µg/mL) or chloramphenicol (25 µg/mL)). *A. tumefaciens* was grown at 28 °C in an LB medium without NaCl and supplemented with spectinomycin (100 mg/mL). *B. cinerea* was grown using 3 different media: (1) MS (malt sporulation medium; 0.5% glucose, 2% malt extract, 0.1% bacto-peptone, 0.1% casein hydrolysate, 0.1% yeast extract, pH 5), (2) MSS (MS supplemented with saccharose 200 g/L (0.6 osmol/L)) and (3) MM2 (minimal medium, as described previously) [31]. Strains were always grown twice on MSS before use. The media used to transform *B. cinerea* are described in Rolland [32] and were supplemented with hygromycin B (75 µg/mL) or neomycin (75 µg/mL) to select the transformants. When needed, the MS medium was supplemented with 0.6 M saccharose (0.6 osmol/L), 0.6 M sorbitol (0.6 osmol/L), 0.6 M KCl (1.2 osmol/L), 0.02% SDS, 300 µg/mL Congo red and 56 mg/mL menadione. Fungal liquid cultures (200 mL MS medium) were inoculated with 10 agar explants (4 mm diameter) and agitated (130 rpm) for 48–72 h at 21 °C. Solid cultures were inoculated with 2 or 4 mm agar explants.

### 2.2. Sclerotial Development, Conidiation and Germination Assays

Sclerotia were observed on MS plates inoculated with agar explants and incubated for 15 days at 21 °C in darkness. Conidia were collected in PBS-tween 0.02% from MS plates inoculated with 10^4^ conidia, incubated in the dark at 21 °C for 3 days and then under near-UV light (365 nm) for 7 days. They were filtered (100 µm) and numerated using a Thoma cell. To assay germination, 10^4^ conidia in 150 µL liquid MS were incubated at 21 °C in a 96-wells plate. During 20 h post-inoculation, images were collected every 30 min under the microscope (×40), and conidia with germ tubes longer than 10 µm were scored as germinated conidia.

### 2.3. Bioinformatic Analyses

To construct the fungal Cps1-like protein family, the Cps1 protein of *C. neoformans* (CNAG_04320) was used as query in the Blastp program [33]. One hundred and fifty-seven different proteomes available on EnsemblFungi (https://fungi.ensembl.org, 1 May 2022) or on the NCBI website (https://blast.ncbi.nlm.nih.gov, 1 May 2022) were searched. Every protein sequence with a significant blast hit and a sufficient coverage (>200 amino-acid) was considered as a Cps1-like protein. Data from the similarity search were used to select the species for the phylogenetic analyses: every *Leotiomyceta* species and 2–4 species for the other taxonomic subphyla or classes were chosen. For the chosen species, every Cps1-like protein was included. Amino-acid sequences were aligned using MAFFT with the L-INS-i algorithm [34]. Regions suitable for phylogenetic inference were selected using BMGE [35]. Default settings were used for MAFFT and BMGE. Maximum likelihood (ML) trees were constructed using PhyML (Guindon et al., 2010) using the following parameters: the LG model with empirical amino-acid frequencies, an estimated proportion of invariable sites, subtree pruning and regrafting (SPR) and five random starting trees added to the standard BioNJ starting tree. A dozen protein sequences likely to disturb the tree topology due to the long branch attraction were excluded (undefined category in Appendix A and Appendix A). The support for each node was estimated using non-parametric bootstraps, with 100 replicates.

The topology of the CPS1 protein was predicted using TOPCONS [36].

### 2.4. Plasmid Constructions and Quantitative Real Time PCR

A Δ*Bccps1* deletion cassette was built following the Fungalbraid (GB) instructions [33]. Briefly, the *Bccps1* 5′ and 3′ flanking regions (FR, 1 kb each) were produced by PCR using genomic DNA as template and primers containing the barcodes GGAG/AATG and GCTT/CGCT, respectively. The amplicons were individually integrated in the pUPD2 plasmid, and the Bsa*I* restriction site located in the 3′ FR was then modified by targeted mutagenesis using IVA cloning [37]. Assemblage to the FB012 bio-brick containing a hygromycin-resistance gene by GB reactions yielded the deletion cassette that was inserted in the pDGB3α1 vector [38]. For compatibility issues with the *A. tumefaciens* LBA1126 strain, the deletion cassette was then transferred to the P7 vector by IVA cloning. The *Bccps1-C* complementation cassette was produced by the amplification of the *Bccps1* CDS flanked by its promoter (1.5 Kb) and terminator (1363 bp) regions. The amplicon was cloned using IVA cloning and the *Pme*I restriction site downstream of the PoliC promoter into a P7 plasmid containing the neomycin resistance gene *nptII*. All expected plasmids were verified by restriction mapping and sequencing before introduction into the *A. tumefaciens LBA1126* strain by transformation.

For quantitative reverse transcription PCR (RT-qPCR), DNA-free total RNA was extracted from lyophilized mycelium using the RNEasy Midi kit (Qiagen, Hilden, Germany) and treated with DNAse using the TURBO DNA-free Kit (Thermofisher, Vilnus, Lituania). 1µg was then used to synthesize the first-strand cDNA with the SuperScript IV First-Strand Synthesis System (Thermofisher, Vilnus, Lituania). RT-qPCR reactions were performed in 96-well plates using ABI-7900 Applied Biosystems (Applied Biosystems, Waltham, MA, USA), using PowerUp SYBR Green Master Mix (Applied Biosystems, Waltham, MA, USA). Relative quantification was based on the 2(^−ΔΔC(T)^) method using the actin (*BcactA*, Bcin16g02020) [39], one elongation factor (*Bcef1α*, Bcin09g05760) and the pyruvate dehydrogenase (*Bcpda1*, Bcin07g01890)-encoding genes as normalization internal controls. Three independent biological replicates were analyzed. All primers used are listed in Appendix A.

### 2.5. Fungal Transformation and Mutant Purification

*B. cinerea* was transformed through ATMT [32], except that frozen conidia (−80 °C, 20% glycerol in water) were used instead of fresh conidia. The transformants were tested by PCR. Genetic purification of the deletion strain relied on single-spores isolation and subcultures on solid MS medium plus hygromycin (75 µg/mL), followed by protoplasts isolation and culture in liquid MSS plus hygromycin (70 µg/mL).

### 2.6. Southern Blot

Genomic DNA was extracted from *B. cinerea* using the DNeasy Plant Mini kit (Qiagen, Hilden, Germany), and was digested by EcoR*I*-HF and PmeI (NEB, Rowley, MA, USA). The 5′ flanking DNA region (720 bp) of the *BcCps1* gene was used as a probe. This DNA fragment was generated by PCR using the For-Cps1-RFG and 3′-southern-Cps1-RFG primers (Appendix A) and the DIG Probe Synthesis Kit (Roche, Mannheim, Germany). The DNA profile produced by hybridization of the labelled probe was revealed by using the DIG Luminescent Detection Kit (Roche, Mannheim, Germany).

### 2.7. Cell Wall Extracts Preparation

The protocol was adapted from [40,41]. Mycelia were collected from 72 h cultures (10 explants in 200 mL of MS liquid, 21 °C, 110 rpm), lyophilized and blended 3 times for 30 s at 30 Hz (TissueLyser II, Qiagen, Hilden, Germany). The resulting cell powder was collected by centrifugation (4000 *g* for 10 min), washed 3 times in 30 mL PBS, suspended in 30 mL PBS-1% SDS, boiled for 15 min, allowed to cool, and collected by centrifugation. The cell wall extracts were washed twice in ice-cold PBS, once in cold water and finally lyophilized and frozen (−80 °C).

### 2.8. Cell Wall Composition Analysis

Cell wall extracts (8 mg) in microtubes were vigorously and horizontally shaken in 1 mL NaOH 0.25 M (30 min). Following a 4-fold dilution in NaOH, 0.2 mg in 100 µL were mixed in TPP plates with 250 µL aniline blue (0.5% in Tris-HCl 0.1 M, pH 9) or Congo red (0.05% in H2O). After rotative agitation (1 h, 500 rpm) in the dark, the plates were centrifuged (5 min, 3000 *g*) to pellet the cell walls and 250 µL of the supernatants were discarded. Water (100 µL) was added, and the cell walls were suspended before fluorescence recording (TECAN Infinite M1000, Ex/Em: 395/500 nm and 530/620 nm, plate cover removed). Labelling with concavalin A-FITC (10 µg/mL, Sigma, Tokyo, Japan) required incubation in Hepes buffer (10 mM pH 7) and 3 washes in water before fluorescence recording (Exc/Em: 490/525 nm).

For enzymatic digestion of cell wall extacts, 5 mg of Lysing enzymes from *Trichoderma harzianum* (Sigma, Tokyo, Japan) or 25 U of lichenase from *Bacillus subtilis* (Megazyme) were dissolved in 300 µL of sodium acetate buffer (50 mM, pH 5) and, respectively, used to digest 1 or 2 mg of cell wall extracts during 3 days at 37 °C. Colorimetric assays were used to quantify the released N-acetyl-glucosamine residues or the reducing sugars [42,43].

### 2.9. Plant Infection and Adhesion Assays 

French bean leaves (*Phaseolus vulgaris* var Saxa) were inoculated with 4 mm mycelial explants or 7.5 µL of conidial suspensions (2 × 10^5^ conidia, collected from MS cultures/mL Gamborg B5 medium (Duchefa, The Netherlands) supplemented with 2% glucose). The leaves were incubated at 21 °C under 80% relative humidity and dark-light (16 h/8 h) conditions.

For adhesion assay, detached bean leaves were inoculated with 4 mm mycelial explants following previous incubation conditions. At 24 hpi, leaves were agitated horizontally in water (8 min, 110 rpm/min), and the detached explants were counted. Adhesion in vitro was assayed in 96 well plates according to Feng et al. 2017 [29]. Conidia (2 × 10^3^ in 250 µL MS) were incubated at 21 °C for 24 h. The medium was discarded and the mycelium was washed 3 times with distilled water and stained for 20 min at room temperature with 130 µL of Crystal violet (0.01% in water). Following 3 washes with water, the mycelium was dried at room temperature and destained with 130 µL acetic acid 30%. OD measurement at 560 nm quantified the cells attached to the wells.

### 2.10. Label-Free Quantitative Proteomics

Exoproteomes were prepared from 72 h cultures (10 explants in 200 mL of MS liquid, 21 °C, 110 rpm). To discard possible fungal matrix polymers, the culture filtrates (11 µM nylon membranes) were frozen in liquid nitrogen, left to thaw at 4 °C overnight and centrifuged at 16,000× *g* for 15 min. The supernatants were mixed with 100% TCA (15% vol/vol) and kept at 4 °C overnight. The protein precipitates were collected by centrifugation (16,000× *g*, 15 min), washed twice in acetone and solubilized in 0.1 M Tris-HCl pH 7, 0.1 M DTT, EDTA 5 mM, glycerol 10%, SDS 4% bromophenol blue 0.008%. Four independent biological replicates were prepared.

Proteins (5 µg) were loaded onto a 10% acrylamide SDS-PAGE gel and visualized by Colloidal Blue staining. Migration was stopped when samples had just entered the resolving gel and the unresolved region of the gel was cut into only one segment. The steps of sample preparation and protein digestion by trypsin were performed as previously described [44]. NanoLC-MS/MS analysis were performed using an Ultimate 3000 RSLC Nano-UPHLC system (Thermo Scientific, Waltham, MA, USA) coupled to a nanospray Orbitrap Fusion™ Lumos™ Tribrid™ Mass Spectrometer (Thermo Fisher Scientific, Waltham, MA, USA). Each peptide extract was loaded onto a 300 µm ID × 5 mm PepMap C18 pre-column (Thermo Scientific, Waltham, MA, USA) at a flow rate of 10 µL/min. After a 3 min desalting step, peptides were separated on a 50 cm EasySpray column (75 µm ID, 2 µm C18 beads, 100 Å pore size, ES903, Thermo Fisher Scientific, Waltham, MA, USA) with a 4–40% linear gradient of solvent B (0.1% formic acid in 80% ACN) for 91 min. The separation flow rate was set at 300 nL/min. The mass spectrometer operated in positive ion mode at a 1.9 kV needle voltage. Data were acquired using Xcalibur 4.4 software in a data-dependent mode. MS scans (m/z 375–1500) were recorded at a resolution of R = 120,000 (m/z 200), a standard AGC target and an injection time in automatic mode, followed by a top speed duty cycle of up to 3 s for MS/MS acquisition. Precursor ions (2 to 7 charge states) were isolated in the quadrupole with a mass window of 1.6 Th and fragmented with HCD@28% normalized collision energy. MS/MS data was acquired in the Orbitrap cell with a resolution of R = 30,000 (m/z 200), a standard AGC target and a maximum injection time in automatic mode. Selected precursors were excluded for 60 s. Protein identification and label-free quantification (LFQ) were performed in Proteome Discoverer 2.5. The MS Amanda 2.0, Sequest HT and Mascot 2.5 algorithms were used for protein identification in batch mode by searching against the ENSEMBL *B. cinerea* ASL83294v1 database (13,749 entries, release 53). Two missed enzyme cleavages were allowed for trypsin. Mass tolerances in MS and MS/MS were set to 10 ppm and 0.02 DA. Oxidation (M) and acetylation (K) were searched as dynamic modifications and carbamidomethylation (C) as static modification. Peptide validation was performed using the Percolator algorithm and only “high confidence” peptides were retained, corresponding to a 1% false discovery rate at peptide level [45]. Minora feature detector node (LFQ) was used along with the feature mapper and precursor ions quantifier. The quantification parameters were selected as follows: (1) unique peptides, (2) precursor abundance based on intensity, (3) normalization mode: total peptide amount, (4) protein abundance calculation: summed abundances, (5) protein ratio calculation: pairwise ratio based, (6) imputation mode: low-abundance resampling and (7) hypothesis test: t-test (background based). Quantitative data were considered for master proteins, quantified by a minimum of 2 unique peptides, a fold changes above 2 and an occurrence in at least 3 of the 4 biological replicates. The mass spectrometry proteomics data have been deposited to the ProteomeXchange Consortium via the PRIDE partner repository with the dataset identifier PXD033590 [46].

### 2.11. Activities of Secreted Enzymes

Proteases, xylanases and laccases activities were measured in De Vallée et al. [47]. Supernatants from the 72 h cultures (10 explants in 200 mL of MS liquid, 21 °C, 110 rpm) were used as enzyme samples. Acid protease activity was measured by incubating (24 h, 30 °C) the samples (150 μL) with 450 μL of 1% hemoglobin (Sigma, Tokyo, Japan), pH 3.5. Reactions were stopped with 25% trichloroacetic acid (400 μL). After centrifugation, supernatants were mixed with 0.5 M NaOH (vol/vol) in UV microplates and optical density was read at 280 nm. Xylanase and cellulase activities were recorded on a plate reader TECAN infinite M1000 using the EnzChek-Ultra-xylanase assay kit (Thermofisher, Vilnus, Lituania) and the cellulase assay kit (Abcam, Cambridge, UK), respectively. Laccase activity was measured by incubating 10 μL of samples and 30 μL of ABTS as substrate (Sigma, Tokyo, Japan) in 230 μL of 50 mM Na-acetate buffer, pH 4.0. Oxidation of ABTS was recorded at 405 nm (Molecular Devices Spectramax-485) during 30 min at 30 °C. One unit was defined as the amount of enzyme producing an increase of 0.01 OD unit per min. Pectin methyl esterase (PME) and amylase activities were measured using plate assays based on Lionetti [48]. 1.5% agarose in 100 mM NaCl was supplemented with 0.1% pectin or 0.1% starch. Wells (5 mm) were punched, filled with samples (40 µL) and the plates were incubated at 37 °C for 24 h. The PME activity was revealed by a dark red circle after flooding of the pectoplates with 5 mL ruthenium red 0.05% (30 min at room temperature) and several washes with water. Similarly, Lugol (20%) was used to reveal the amylase activity as clear circular halos in the starch plates.

## 3. Results

### 3.1. The Botrytis Cps-Encoding Genes

Cps proteins constitute a fungal family of type two glycosyltransferases (GT2, Figure 1A). In the species from the *Leotiomyceta* order, this family is split into two paralog subfamilies, Cps1 and Cps2, due to a gene duplication in their ancestor genome (Appendix A) [28]. A member of the Cps1 subfamily was clearly identified in the 39 ascomycetous *Leotiomyceta* species analyzed (Appendix A, Appendix A). In contrast, 14 of these 39 species no longer carry a gene of the Cps2 subfamily, including *Sclerotinia sclerotorium* and *Blumeria graminis*, the proximate species of *B. cinerea* (Appendix A), while the latter does carry a *cps2* gene. Moreover, the amino-acid sequences of the Cps2 proteins are less conserved than that of the Cps1 proteins, as shown by their longer branches in the phylogenetic tree (Appendix A). In *B. cinerea*, the expression of *Bccps1* (Bcin05g06910) has been detected during the germination of conidia [49] and during vegetative growth in liquid cultures, as well as in solid cultures enriched in infection cushions, where it is moderately (×2.7) increased [13]. On the other hand, the expression of *Bccps2* (Bcin03g03750) has never been observed in the transcriptomic analyses conducted so far. Together, the difference in gene conservation and in gene expression suggests a likely functional role for *Bccps1* and a secondary or dispensable role for *Bccps2*. However, our attempt to isolate a homocaryotic ∆*Bccps1* mutant line was successful, whereas the isolation of a homocaryotic ∆*Bccps2* mutant line resisted 10 rounds of single-conidia or protoplasts purification steps. In the absence of a *Bccps2* mutant line, this study focuses on the characterization of the *Bccps1* gene.

Cps1 proteins share a common topology of integral membrane protein with the conserved catalytic GT2 domain located in the central region of the protein (Figure 1B). When submitting the sequence of BcCps1 to the algorithm AlphaFold Monomer v2.0, a 3D model of the protein was obtained (Figure 1C) that was very similar to the only two other models available for Cps1, from the plant pathogen *Z. tritici* (https://alphafold.com/entry/F9WWD1, 1 May 2022) and the human pathogen *Cladophialophora carrionii* (https://alphafold.com/entry/A0A1C1CM71, 1 May 2022). These models highlight a membrane anchoring of Cps1 through one N- and two C-terminal transmembrane (TM) helices. The catalytic GT2 domain is positioned on the cytoplasmic face of the membrane, as well as the C-terminal tail of the protein that contains the conserved W motif and ends with a disordered region. Similarly to the situation in chitin synthases [11], one hydrophobic helix is predicted not to cross the membrane. The model proposes its longitudinal orientation in the membrane (Figure 1C).

### 3.2. Deletion of the Bccps1 Gene and Localization of the BcCps1 Protein

A hygromycin B resistance cassette (Appendix A) was introduced in *B. cinerea* via *A. tumefaciens*-mediated transformation (ATMT) to replace the *Bccps1* gene. The selected transformants were heterocaryons. Based on the work of King et al. [28], homocaryons were sought by isolating protoplasts prepared from these transformants in a rich liquid medium supplemented with saccharose (an osmo-protective agent). After two successive rounds of this purification process, PCR diagnostics and Southern blot analysis confirmed the isolation of the expected ∆*Bccps1* deletion strain by showing the absence of the parental *Bccps1* locus and the presence of the unique, expected DNA fragment corresponding to the mutated locus (Appendix A). Subsequently, a complemented strain was obtained by an ATMT-mediated random insertion of a DNA cassette carrying the *Bccps1* gene (Appendix A) inside the genome of the deletion mutant.

To get a clue about the localization of the BcCps1 protein, transformed lines of *B. cinerea* were constructed that carried a DNA cassette coding for a BcCps1 protein fused to a green fluorescent protein (GFP, optimized for *B. cinerea* [50]). The GFP sequence was inserted in 3 different positions of the BcCps1 sequence: (1) at the N-terminus, (2) after the predicted N-ter TM helix and (3) at the C-terminus (Appendix A). Unfortunately, a fluorescent signal could only be detected in transformants producing BcCps1 fused to GFP at the C-terminus, and this signal was intra-vacuolar (Appendix A). Such labelling, identical to that reported for Cps1 in *M. oryzae* and in *A. nidulans* [24,29], suggests the cleavage of the fusion protein while the absence of GFP signal in the other transformants suggests folding issues not compatible with the production of the recombinant proteins.

### 3.3. Role of BcCps1 in Mycelial Growth

When the Δ*Bccps1* mutant strain was inoculated on a rich solid medium, it grew 90% slower than the parental strain (Figure 2A), and a similar result (83% reduction) was obtained on a minimal medium (data not shown). In a rich liquid medium, the Δ*Bccps1* mutant produced 65% less biomass than the parental strain (Figure 2B), indicating a strong growth impairment under these conditions as well, but less than on a solid medium. Moreover, the liquid cultures revealed a small pellet phenotype in the mutant strain. In comparison with the uniform spongy pellets produced by the explants of the parental strain used to inoculate the medium, more numerous and smaller pellets were collected from the development of the Δ*Bccps1* explants that appeared smoother, denser and drier (Figure 2C). These results indicate that BcCps1 plays an important role in vegetative growth. In the complemented strain, the parental growth rate and biomass production were restored (Figure 2), and this argued for the Δ*Bccps1* mutation being solely responsible for the observed growth phenotypes.

The difference in pellet size and number suggested an increased fragility of the mutant hyphae that would result in mycelium-breaking events during growth under agitation and the subsequent rise of satellite pellets. Interestingly, when the mutant hyphae grew out of a droplet of liquid medium onto a plastic surface, frequent hyphal rupture events were revealed by the leakage of the cell content at the breaking sites (Figure 3A). As to the aspect of the Δ*Bccps1* mycelial pellets, it was suggestive of a modified cell wall or extra-cellular matrix and/or a modified mycelium development. In relation with this, microscopy analysis of the mutant hyphae expanding from conidia on agar plates revealed no difference when compared with the parental strain during the first 24 h of growth, but a hyperbranched mycelium latter on (Figure 3B). On a plastic surface, again no difference could be observed in the first 24 h between the mutant and parental strains, but tufts of ramifications (multibranching) at the margin of the nascent colony were then observed in the mutant, and not in the parental, strain (Figure 3B). To better evaluate how these changes in mycelium development over time could relate to the *Bccps1* gene, the expression of that gene was monitored in the parental strain at 24, 48 and 72 h of growth. As shown in Figure 3C, a 10-fold increase in *Bccps1* expression was recorded between 24 h and 48 h, and this increased almost 5-fold more between 48 h and 72 h. Altogether, these results suggest that the conidia of *B. cinerea* give rise to a mycelial colony through two phases, and that *Bccps1* plays a role in mycelium development and integrity during the second phase. In the complemented strain, hyphal rupture events were not seen and the parental branching pattern of the mycelium was restored.

### 3.4. Role of BcCps1 in Sclerotia Formation, in Conidiation and in Germination

Depending on the environmental conditions, the mycelia of *B. cinerea* can produce resistant structures called sclerotia or conidiophores that generate asexual spores (conidia). To explore the role that BcCps1 might play in these developmental processes, explants of the parental and mutant strains were used to inoculate plates of rich medium that were incubated in the dark for 2 weeks in order to favor the formation of sclerotia (Figure 4A). As expected, the parental strain produced numerous white sclerotial initials that matured into melanized (dark) and dry sclerotia at 12 to 15 days of culture. In comparison, the mutant strain produced white and dense cotton-like masses that resembled the sclerotial precursors produced by the parental strain at day 9, but these structures never turned into sclerotia, even when the incubation time was prolonged (up to 4 weeks, data not shown). This demonstrates that BcCps1 is involved in the process underlying sclerotial development, and rather during maturation than during the first stages of their development.

A second set of plates was incubated under near-UV illumination to favor asexual reproduction. In comparison with the parental colonies, the mutant mycelium produced colonies with powdery patches (Figure 4B). Counting of the conidia recovered from these plates showed no significant reduction in sporulation in the mutant strain, but microscopy observations revealed that 43% of the mutant’s conidia were deformed, a proportion 13 times superior to that found in the parental sample (Figure 4C). This suggests that BcCps1 is involved in the process associated with asexual spore formation. By contrast, it does not participate in the conidial germination process, as the counting of conidia producing a germ tube showed no significant difference between the parental and mutant strains, including the deformed conidia that also germinated (Appendix A). Lastly, in the mutant strain, only microconidia were noticed alongside the collected conidia. In the complemented strain, the aspect of the reproductive mycelium was fully restored (Figure 4B) and the default in conidia formation was corrected by 60% (Figure 4C). On the other hand, the default in sclerotia formation was not corrected (Figure 4A).

### 3.5. Role of BcCps1 in Cell Wall Integrity and Composition

The integrity of the Δ*Bccps1* mutant cell wall was analyzed by growing the mutant mycelium on solid media containing osmotic, oxidative or cell wall-stressing agents. When compared to the parental strain, this revealed an 28% increased sensitivity of the mutant strain to Congo red (Figure 5A), while a similar tolerance to hyperosmotic conditions (KCl), to SDS and to menadione was recorded (Appendix A). In parallel, the effect of osmoprotective molecules such as sorbitol and saccharose on the growth of the mutant strain was explored. Whereas these molecules had no effect on the growth of the parental or complemented strains, they, respectively, showed a 151% and 188% positive effect on the growth of the mutant mycelium (Figure 5B).

To gain information about the cell wall composition of the ∆*Bccps1* mutant, miniaturized fluorescent assays were developed to analyze cell wall extracts (Figure 5C). The first marker to be selected was aniline blue for its known interaction with β-1,3 glucans [51]. Upon mixing with the mutant cell wall extracts, it generated a fluorescent signal that was half (-48%) of the signal detected with the cell extracts of the parental strain. Concanavalin-A was then selected for its established interaction with glycoproteins [52]. The interaction of a fluorescent derivative (FITC) of this molecule with the mutant cell wall extracts produced a signal 37% less intense than the one produced with the parent counterpart. Finally, Congo red was selected for its reported broader spectrum of interactants, namely chitin, some glucans and amyloid proteins [53]. To this molecule the mutant cell wall extracts interacted one-third less (−32%) than the control extracts. Since Congo red can interact with chitin, this last result questioned a possible chitin content reduction in the mutant cell wall. The fluorescent chitin marker calcofluor was therefore introduced in the miniaturized assay [54], but its strong intrinsic fluorescence forced washing steps that caused too much variation across the replicates to achieve consistent readings. As an alternative, the parent and mutant cell wall extracts were treated with cell wall lysing enzymes and the release of N-acetyl-glucosamine residues was quantified, showing no difference between the two strains (Appendix A). Lastly, the cell wall extracts were digested by a lichenase to detect the putative presence of β-1,3-1,4 glucan in *B. cinerea* and to test the role suggested in *N. crassa* for Cps1 in the biosynthesis of this polysaccharide [30]. Products of the enzyme reaction were detected, arguing for the presence of this polymer in the cell wall of *B. cinerea*, but no difference could be observed between the mutant and parental strains (Appendix A). Altogether, these results suggest a modified integrity and composition of the cell wall in the mutant strain, with a decrease in β-1,3 glucans and glycoproteins contents and probably no change in chitin or β-1,3-1,4 glucan contents. Re-introduction of the *Bccps1* gene in the mutant strain partially restored its cell wall composition and integrity (Figure 5).

### 3.6. Exoproteome Analysis of the ΔBccps1 Mutant Strain

The exoproteome of the mutant strain was explored through a shotgun proteomic analysis and compared to that of the parental strain. Both strains were grown in liquid cultures under the same conditions used to prepare cell wall extracts. Four biological replicates were prepared, and the proteins present in the supernatants were precipitated and subjected to mass spectrometry analysis. The proteins identified with a minimum of two unique peptides, an occurrence of at least three biological replicates out of four and a fold change above two between the two strains were retained for the comparative evaluation of the effect of the mutation on the exoproteome. This resulted in the listing of 132 proteins up-accumulated and 51 proteins down-accumulated in the mutant exoproteome when compared with the parental counterpart (Appendix A). Based on the predicted presence of signal peptides (SP) [55] and TM, 157 of these proteins (85.8%) would be secreted through the conventional secretory pathway. Further *in silico* analysis, including manually curated *B. cinerea* protein lists [13], led to a functional classification for 140 of the 183 proteins (Appendix A). As shown in Figure 6A, the mutant exoproteome is characterized by the up-accumulation of FCW proteins, of enzymes involved in metabolism and in oxidoreduction and of proteases, cellulases and proteins related to nucleic acids metabolism. More detailed analysis distinguished sub-categories of metabolic enzymes (related to lipids/sterols and to phosphate) and sub-categories of FCW proteins (Figure 6B). The first FCW sub-category relates to chitin and contains two chitinases and three chitin deacetylases. The second sub-category relates to α glucan and contains three glycosyl hydrolases (family 71). The third sub-category relates to β glucan and contains nine proteins, among which are BcGas5, BcBgl2 and BcScw11. The last sub-category contains two glycoproteins and three proteins involved in the cross linking of chitin and glucans, including the cell death-inducing protein BcCrh1 [56]. The last proteins whose accumulation in the exoproteome were modified by the Δ*Bccps1* mutation were plant cuticle- and plant cell wall-degrading enzymes (PCWDE), among which were pectinases, hemicellulases and amylases (Appendix A). However, the accumulation profile was mixed for these proteins, with some being up-accumulated and others being down-accumulated (Figure 6A).

To complete these data, biochemical assays were performed to measure enzymatic activities. The parental and mutant supernatants used to prepare the exoproteome samples were confronted to substrates specific of proteases, xylanases, laccases, amylases, pectinase/pectin-methyl-esterases (PME) and cellulases. In consistence with the up-accumulation of proteases in the mutant exoproteome, more proteolytic activity (200%) was measured in the mutant secretome (Figure 6C). A reduced amylolytic activity (−95%) also corroborated the down-accumulation of two amylases in this secretome (classified in Other CAZy in Figure 6A). On the other hand, the mixed up- and down-accumulations of laccases (three up, two down, classified in Oxidoreduction in Figure 6A), xylanases (three each) and PMEs (one each) in the mutant exoproteome were associated with a 90%, 68% and 98% reduction in laccase, xylanase and PME activity, respectively. Under our conditions, no cellulase or pectinase activity was detected in either strain. Apart from the proteases, these results revealed a weak enzymatic arsenal in the Δ*Bccps1* secretome that was restored to the parental level in the complemented strain (Appendix A).

### 3.7. Role of BcCps1 in the Interaction of B. cinerea with a Host Plant

To evaluate the impact of the *Bccps1* deletion on the virulence of *B. cinerea*, both mycelial explants and conidial suspensions were used to inoculate bean leaves. When the experiment was performed with explants, the parental hyphae rapidly colonized the plant tissues to produce clearly visible necrosis areas at 2 dpi (Figure 7A). At 4 dpi, the leaves were half colonized. In the case of the mutant strain, small areas of necrosis could be observed, but this was delayed by at least 2 days when compared with the parental strain and showed no evolution with time (Figure 7A). In addition, 40% of the mutant’s explants triggered no necrosis (Figure 7B) and 40% of the mutant’s explants were defective in adhering to the leaf surface (Figure 7C). These results indicate that BcCps1 plays a role in hyphal adhesion to a plant surface and in tissue invasion.

When conidial suspensions were used in the infection assays, the mutant strain triggered a visible plant reaction similar to that observed with the parental strain at 2 dpi. However, the parental mycelium then colonized the leaves, while the mutant mycelium did not (Figure 7D). This questioned the capacity of the mutant conidia at germinating on the plant surface, the capacity of the mutant mycelium at adhering to this surface and/or the capacity of the mutant mycelium at producing infection cushions. Microscopy analysis at 2 dpi revealed that the mutant conidia had germinated and had produced mycelium at the plant surface, even more so than the parental strain (Figure 7E). In parallel, adhesion assays *in vitro* showed that the mycelium produced in 24 h by the conidia of the mutant strain was not affected in adherence. On the other hand, in 48 h of hyphal development on either glass or plastic surfaces, this mycelium produced numerous hooks (precursors of infection cushions (IC)), but rare ICs when compared with the parental mycelium (Figure 7F). This suggests that BcCps1 plays a role in the development of ICs, and therefore in the penetration step of plant infection. Re-introduction of the *Bccps1* gene in the mutant strain restored the capacity of the mycelium at adhering to bean leaves and at forming IC in vitro (Figure 7C,F). It also restored the capacity of the fungus at colonizing leaves, albeit partially (Figure 7A,D).

## 4. Discussion

Cps1-encoding genes are present in a large number of fungal species of the phyla *Basidiomycota, Ascomycota*, *Mucoromycota*, *Zoopagomycota* and *ChitridiomycotA.* However, no ortholog has been detected in *Cryptomycota* or *Blastocladiomycota* and, even in the fungal phyla where the gene is present, some species have presumably lost it. In ascomycetes, this is for example the case for the *Taphrinomycotina*, the *Saccharomycotina*, the *Pezizomycetes* and for some *Orbiliomycetes* species (Appendix A; Appendix A) [28]. On the contrary, all studied species of the *Leotiomyceta* order have at least one ortholog and sometimes more, due to duplications. One such duplication occurred in the common ancestor of *Leotiomyceta* [28] and gave rise to the Cps1 and the Cps2 paralog subfamilies (Appendix A). The Cps1 subfamily is the most conserved, and its member in *B. cinerea* was functionally characterized in this study. As in the other Cps1 proteins already studied in filamentous ascomycetous fungi [22,23,24,25,26,27,28,29,30], the BcCps1 protein belongs to the glycosyl transferase type 2 polysaccharide synthase family (GT2) and is supposed to contribute to the biology of the cell wall in *B. cinerea*.

### 4.1. BcCps1 Is Required for Normal Composition and Integrity of the Fungal Cell Wall

Fungal cell walls are all described as containing the well-known polysaccharides chitin and β-1,3 glucan. In some species, the presence of other glucans has also been demonstrated (β-1,6; α-1,3; mixed β-1,3/β-1,4) [57] while, in addition, yet unidentified polysaccharides are expected. An important role attributed to some of these polymers relates to the integrity of the cell wall and in the resistance of fungal cells to environmental stress. Here, we showed that the deletion of the *Bccps1* gene associates with cell rupture events and the emptying of the cellular content when hyphae grow out of a medium droplet onto a solid surface. We also showed that the ∆*Bccps1* mutant strain produces numerous satellite pellets in agitated liquid cultures, strongly suggesting hyphal fragmentation under these conditions. Further analysis revealed an increased sensitivity of the ∆*Bccps1* strain to Congo red (cell wall stress) and a positive effect of saccharose, or sorbitol (osmoprotective agents), on its growth in a rich solid medium. At the biochemical level, the mutant’s cell wall is characterized by a reduction in β-1,3 glucan. Altogether, this indicates that the BcCps1 protein is required for normal polysaccharidic composition and normal integrity of the cell wall in *B. cinerea.* Of note, the absence of BcCps1 seems to affect neither the cell sensitivity to hyperosmotic stress, nor the cell wall content in chitin or β-1,3-1,4 glucan (lichenan). In consistence with various profiles of cell wall polysaccharides contents in *cps1* mutants of *N. crassa*, *A. nidulans* and *A. fumigatus* (Figure 1), all polysaccharides of the *B. cinerea* cell wall are therefore not disturbed in the absence of BcCps1. Concerning lichenan, its presence has not yet been reported in the cell wall of *B. cinerea* to our knowledge, and our results suggest that it is not produced by BcCps1. This contradicts the proposition of Fu et al. [30] but it would be consistent with the fact that the ortholog of the lichenan synthase identified in *A. fumigatus* is not *Bccps1,* but rather another gene (Bcin04g06040) in *B. cinerea* [5].

Fungal cell walls also contain proteins. Many of them are glycoproteins, and some are known to associate to β-1,3 glucan (Pir proteins) or to to β-1,6 glucan (GPI-anchored proteins) [57,58]. In the cell walls of ∆*cps1* mutants in *A. nidulans* and *N. crassa*, the amount of glycoproteins is reduced (Figure 1). Furthermore, more proteins are detected in the surrounding medium of the *N. crassa* ∆*cps1* mutant [30]. Here, we showed that 132 proteins up-accumulate in the exoproteome of the ∆*Bccps1* mutant, while less glycoproteins (−37%) are found in its cell wall. These data suggest that proteins may not incorporate properly into the wall matrix in the absence of Cps1. Combined with the above-described changes in the polysaccharidic composition of the FCW, this makes Cps1 a protein necessary for the full construction of this structure. 

### 4.2. BcCps1 Is Involved in Hyphal Adherence and Reveals a Two-Phase Hyphal Growth on Solid Surfaces

In all ascomycetous *cps1* mutants studied so far, the radial growth of the colony is impaired (Figure 1). In *B. cinerea*, not only this is confirmed, but the reduction is the strongest (−90%) up to now reported. At the microscopic level, mutant conidia germinate normally on a plastic or agar surface in vitro and give rise to a young mycelium that develops with no apparent default during 24 h. Then, abnormal tufts of ramification (multibranching) or hyperbranching events occur within the mycelium, and hyphal expansion drops drastically. The deletion of the *Bccps1* gene hence reveals that hyphal growth in *B. cinerea* seems to proceed in two phases on a solid surface after conidia germination: one initial Cps1-independent phase and one subsequent Cps1-dependent expansion phase. Interestingly, an increased expression of *Bccps1* was measured between 24 h, 48 h and 72 h of growth, and this suggests that BcCps1 could play an important role in the unfolding of the second phase of hyphal development in *B. cinerea.* As proposed by King et al. [28], Cps1 could produce a cell wall (or extracellular matrix) polysaccharide necessary for hyphal growth on solid matrices.

In *A. nidulans* and *A. fumigatus*, a decreased adherence capacity has been documented in *cps1* mutants (Figure 1). Interestingly, young mycelia produced in 24 h by conidia of the ∆*Bccps1* strain exhibited no adhesion default, while explants of the mutant mycelium showed a 40% loss in adhesion to leaves (and also to plastic surfaces, data not shown). A conserved role for Cps1 in mycelial adherence might therefore be considered. Cps1 could produce a polysaccharide that directly participates in hyphal adherence, as established for α-1,3 glucan and galactosaminogalactan [59,60].

The growth of the ∆*Bccps1* mutant strain is also altered in liquid cultures, but less than in solid media (−65%). Together with the reduced growth of the *A. nidulans* ∆*cps1* mutant in liquid cultures [27], this indicates that Cps1 also plays a role in hyphal development in liquid environments. Furthermore, the liquid cultures revealed a small pellet phenotype in the ∆*Bccps1* mutant that corresponds to pellets more compact, smaller and smoother than their parental counterparts. A similar phenotype is observed in the ∆*cps1* mutant of the rice pathogen *M. oryzae* [24], and this raises the possibility of a conserved function of Cps1 during hyphal growth in liquids. Remarkably, a similar phenotype also characterizes mutants of *A. nidulans* and *A. oryzae* affected in α-1,3 glucan and/or galactosaminogalactan synthesis, the above-mentioned polysaccharides involved in surface adherence, but also involved in hyphal aggregation (self-adhesion) in liquid cultures [60]. If the ∆*cps1* mutation caused a reduction or a mislocalization of these polymers in the FCW, this could explain the adherence and aggregation defects documented in *B. cinerea*, *A. nidulans*, *A. fumigatus* and *M. oryzae*. It could also be consistent with the strong up-regulation of the α-1,3 glucan synthase-encoding gene in the ∆*cps1* mutant of *F. graminearum* [28] as a compensatory mechanism.

### 4.3. BcCps1 Is Involved in Conidial and Sclerotial Differentiation

Under favorable conditions, including exposition to light, the life cycle of *B. cinerea* ends with asexual reproduction and the dispersal of conidia propagating the fungus and the disease. Alternatively, the mycelium can differentiate sclerotia for long-period survival in soils and potential sexual reproduction. In the absence of Cps1, the production of conidia is abolished in *M. oryzae* and reduced in *N. crassa*, as well as in *A. nidulans* and *A. fumigatus* (Figure 1). Deletion of *Bccps1* in *B. cinerea* does not impair the production of conidia, but it increases the percentage of conidia with abnormal shapes. Considering that fungal cells are mainly shaped by their cell wall, this could relate to the already discussed role of BcCps1 in cell wall composition/integrity.

Regarding the differentiation of sclerotia, BcCps1 plays a critical role, and it is noticeable that its ortholog in *N. crassa* plays a crucial role in protoperithecia differentiation [30]. BcCps1 is not essential to the aggregation and tangling of the aerial hyphae that form the sclerotial initials. It is not essential either to the subsequent hyphal growth and aggregation that shape and enlarge the sclerotia. However, its absence stops the maturation process before the melanization and densification of the shell. The loss of laccase activity in the mutant’s secretome could partly account for this, as these enzymes are potential candidates in the melanization process [61]. Furthermore, due to the melanin granules being cross-linked to the cell wall polysaccharides [62], the role of BcCps1 in the maturation of sclerotia may again be linked to its role in the biology of the FCW. 

### 4.4. BcCps1 Is Involved in Virulence

The loss of the BcCps1 protein associates with an incapacity of the ∆*Bccps1* mutant conidia at infecting plants. In this respect, our study cumulates with the previous studies of fungal ∆*cps1* mutants that report full or partial loss of virulence in five pathogenic species (Figure 1). The mutant conidia germinate into a young mycelium at the plant surface, but fail to initiate infection. This could be explained by the negative impact of the *Bccps1* deletion on the production of infection cushions, organs involved in plant penetration and necrosis [13]. By contrast, some ∆*Bccps1* mycelial explants can trigger, although with delay, small lesions of the plant tissues. Considering the growth defect of this mycelium, its increased fragility and sensitivity, and its reduced secretion of PCWDE, this residual capacity at resisting the defense of the host and at colonizing its tissues was unexpected. The increased secretion of some proteases and oxidoreductases might account for this.

## 5. Conclusions

Our study of *Bccps1* in *B. cinerea* identifies this gene as an important player in the biology and life cycle of this pathogen. This gene being absent in plants, mammals and insects makes the BcCps1 protein a putative target for grey mold disease control. Its absence affects the FCW, and causes pleiotropic effects as a possible consequence. Identification of the putative polysaccharide produced by this GT2 enzyme would now be of great interest.

## Figures and Tables

**Figure 1 jof-08-00899-f001:**
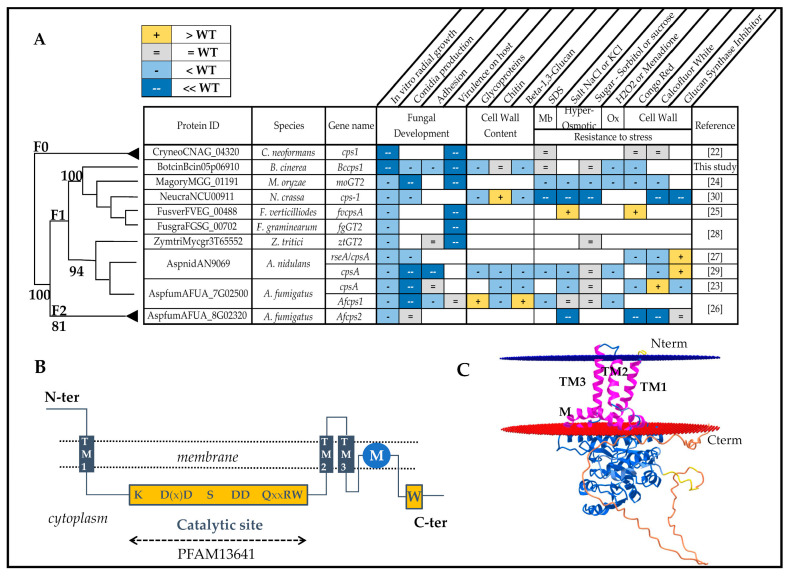
BcCps1 and its characterized homologs. (**A**) Presentation of *cps1* genes already studied in filamentous fungi and of the Δ*cps1* mutant phenotypes. The simplified phylogeny is extracted from that of the Cps1 protein family (Appendix A). Membrane (Mb), Oxydative (ox). (**B**) Predicted secondary structure of BcCps1 inside a membrane (dotted line), showing the three TM helices, the catalytic and W domains with the conserved amino-acids (capital letters) and the non-TM hydrophobic helix (M). (**C**) Predicted 3D model of the *B. cinerea* BcCps1 protein within a lipid bilayer (blue and red tilted circles). The structure was predicted using AlphaFold Monomer v2.0 and the 3D representation was obtained using the OPM prediction tool (https://opm.phar.umich.edu/ppm_server2, 1 May 2022). The TM helices and the non-TM hydrophobic helix are colored in pink. The other protein fragments are colored following the AlphaFold confidence code (blue (high), yellow (medium), orange (low)).

**Figure 2 jof-08-00899-f002:**
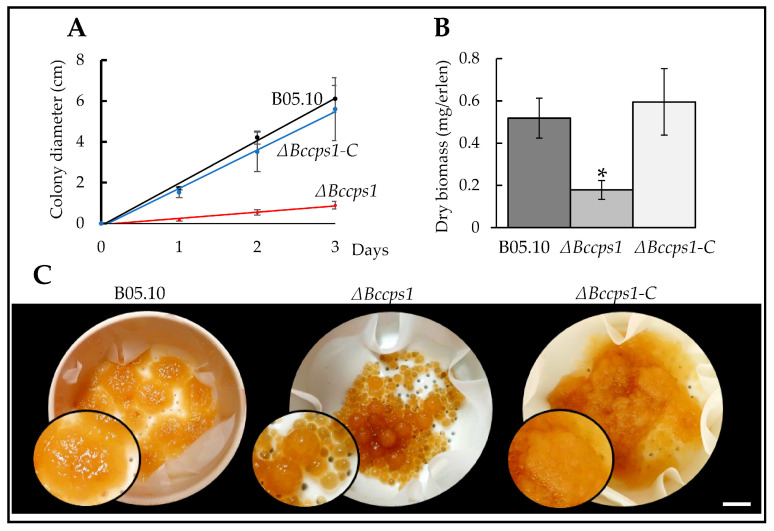
BcCps1 is involved in mycelial growth. (**A**) Radial growth of the parental (B05.10), mutant (Δ*Bccps1*) and complemented (Δ*Bccps1*-C) strains on a solid rich medium over time. (**B**) Growth of the parental, mutant and complemented strains in a liquid rich medium. The biomass was weighed at day 3. (**C**) Images of the mycelial pellets collected by filtration of the liquid cultures at day 3. Black dots among the pellets are the funnel holes visible through the wet filter paper. Magnifications are shown in black circles. Scale bar: 1 cm. Stars indicate significant differences (*T*-test bilateral; *p* < 0.01).

**Figure 3 jof-08-00899-f003:**
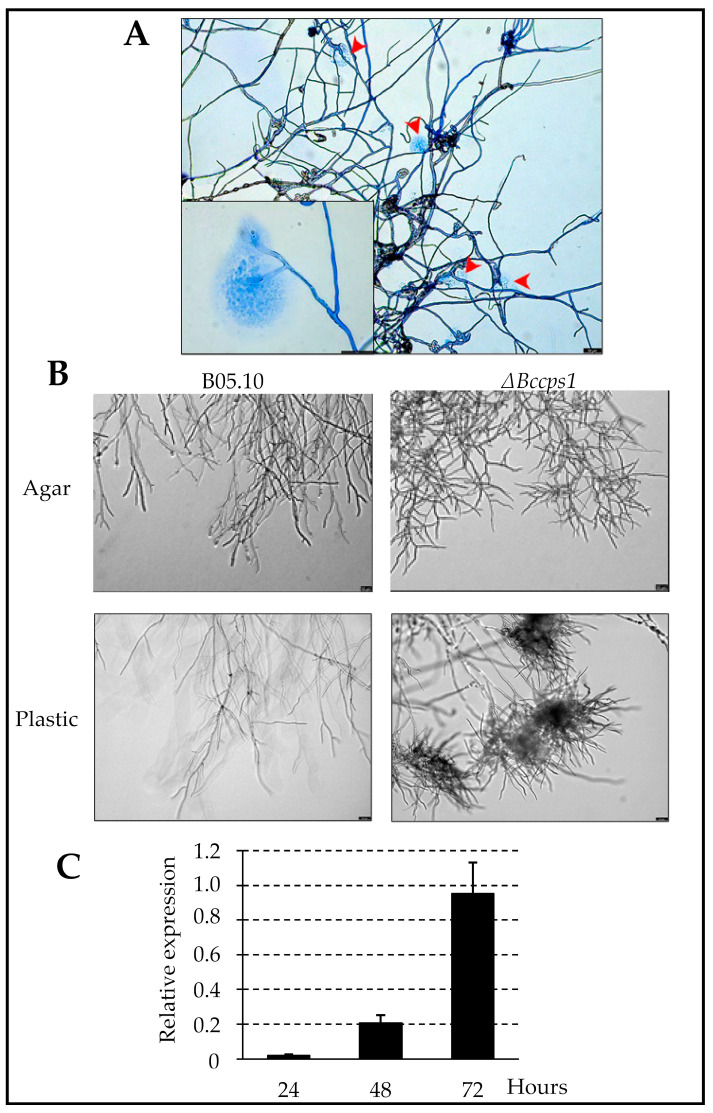
BcCps1 is involved in hyphal development 24 h after conidia germination. (**A**) Rupture and cell content leakage events (red arrows) in the hyphae of the mutant strain growing out of medium droplets on a plastic surface. Hyphae were stained with cotton blue. Scale bar: 50 nm—inlet: higher magnification. (**B**) Parental (B05.10) and mutant (Δ*Bccps1*) hyphal development following conidia germination (after 24 h) on a rich solid medium (Agar) and inside a droplet of rich medium on a plastic surface (Plastic). Scale bar: 50 nm. (**C**) Expression kinetic of the *Bccps1* gene (relative to the expression of three reference genes (*BcactA*, *Bcef1α* and *Bcpda1*) in the parental strain grown for 72 h in a rich liquid medium. Means and standard deviations were calculated based on three independent experiments (*n* = 9).

**Figure 4 jof-08-00899-f004:**
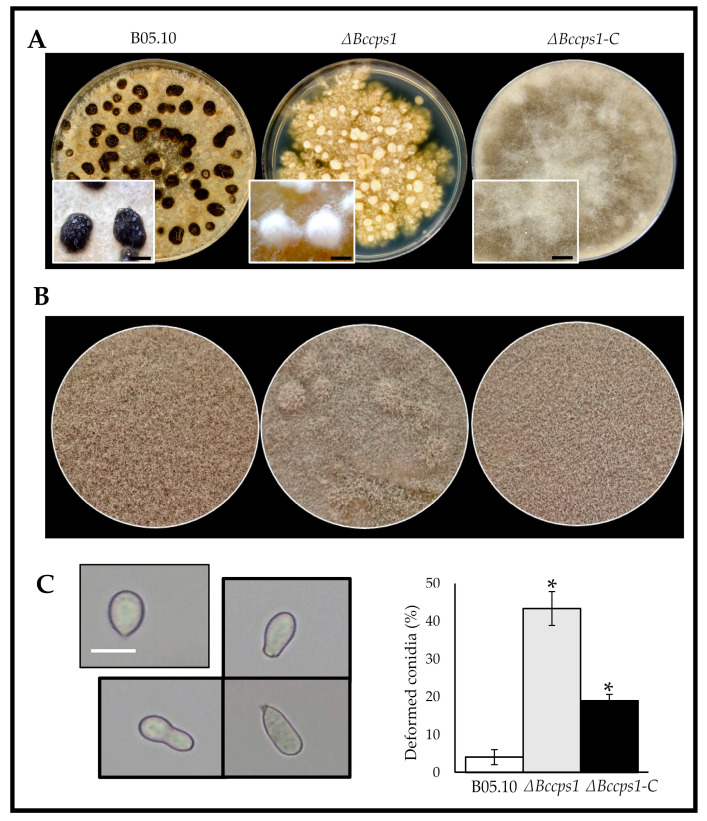
BcCps1 is involved in sclerotia and conidia formation. (**A**) Sclerotial development in the parental (B05.10), mutant (Δ*Bccps1*) and complemented (Δ*Bccps1*-C) strains grown in darkness for 15 days. Dark sclerotia are visible at the surface of the parent mycelium covering the culture plate, while the mutant colony produces white cotton-like masses. Magnifications of both structures are shown in rectangles. Scale bar: 0.5 cm. (**B**) Images of sporulating plates of the parental, mutant and complemented strains grown under near-UV light. (**C**) Images (left) and counting (right) of the usual apiculate (top left) and deformed (black frame) conidia produced by the parental, mutant and complemented strains (Size bar = 10 µm). Means and standard deviations were calculated from three independent experiments (*n* = 150). Stars indicate significant differences (*T*-test bilateral; *p* < 0.01).

**Figure 5 jof-08-00899-f005:**
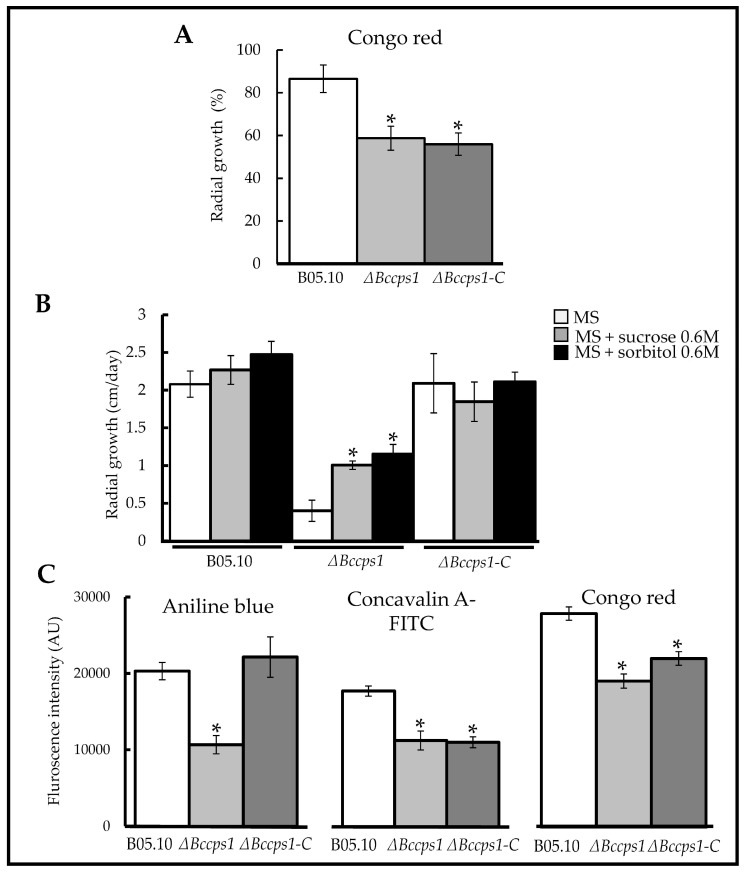
BcCps1 is involved in cell wall integrity and influences cell wall composition. (**A**) Sensitivity of mycelial growth to Congo red in the parental (B05.10), mutant (Δ*Bccps1*) and complemented (Δ*Bccps1*-C) strains. 100% corresponds to a radial growth of 2.2 cm per day. (**B**) Impact of 0.6 M saccharose (grey) and 0.6 M sorbitol (black) on the mycelial expansion of the B05.10, Δ*Bccps1* and Δ*Bccps1*-C strains inoculated onto a rich medium (white). (**C**) Cell walls reactivity to anilin blue (β-1,3-glucan marker), ConcavalinA-FITC (glycoproteins marker) and Congo red (marker of chitin, some glucans and amyloid proteins) in the parental, mutant and complemented strains. Means and standard deviations were calculated from three (**A**,**B**) or six (**C**) independent experiments (*n* = 9 (**A**,**B**) or *n* = 17 (**C**)). Stars indicate significant differences (*T*-test bilateral; *p* < 0.01).

**Figure 6 jof-08-00899-f006:**
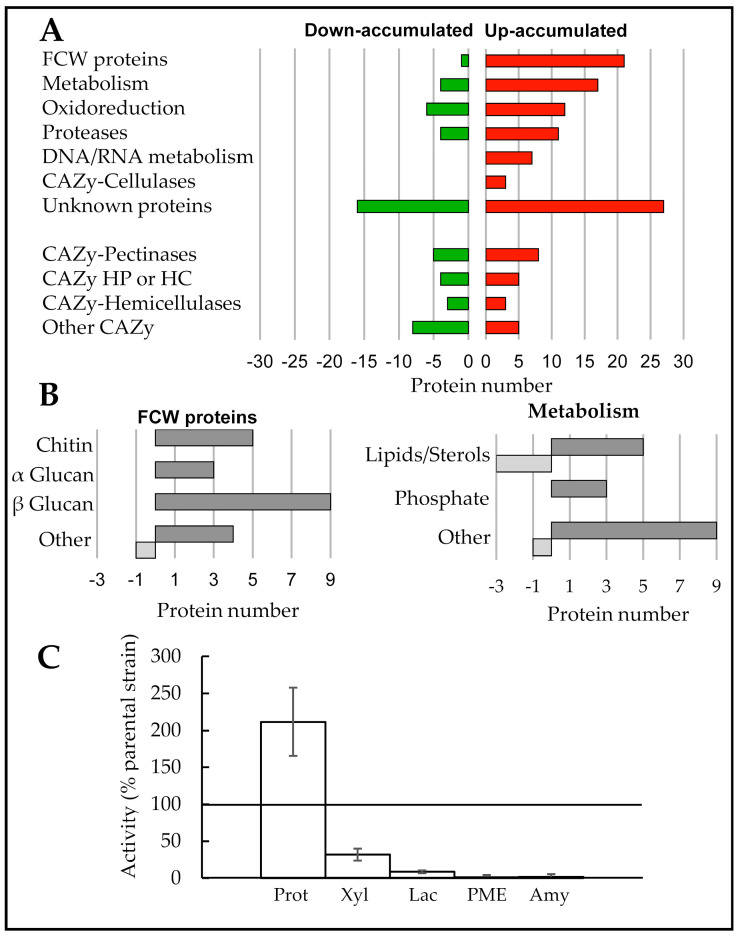
BcCps1 influences the fungal exoproteome. (**A**) Comparative shotgun analysis of the proteins present in the supernatants of 3-days liquid cultures of the parental and Δ*Bccps1* mutant strains. The proteins down- or up-accumulated in the mutant exoproteome are classified according to their functional category (fold change ≥ two, occurrence ≥ three of four biological replicates, see Appendix A). CAZy, carbohydrate active enzymes; FCW, fungal cell wall; HP, hemicellulose pectin; HC, hemicellulose cellulose. (**B**) Sub-categorization of the categories « FCW proteins » and « Metabolism » (see Appendix A). The down-accumulated proteins (green) are plotted as negative numbers. (**C**) Enzymatic activities measured on the same supernatants used for the proteomic analysis (four replicates). The results are shown as percentages of the parental strain values. PMEs; pectin methyl esterases. Means and standard deviations were calculated from three independent experiments (*n* = 12). 100% activity in the parental strain correspond to 0.41 OD^280^/mg dry mycelium (Proteases), 36 RFU/min.mg dry mycelium (Xylanases), 7.9 × 10^−3^ RFU/min.mg dry mycelium (Laccases), 2.35 cm/40 µL supernatant (PMEs) and 2 cm/40 µL supernatant (Amylases).

**Figure 7 jof-08-00899-f007:**
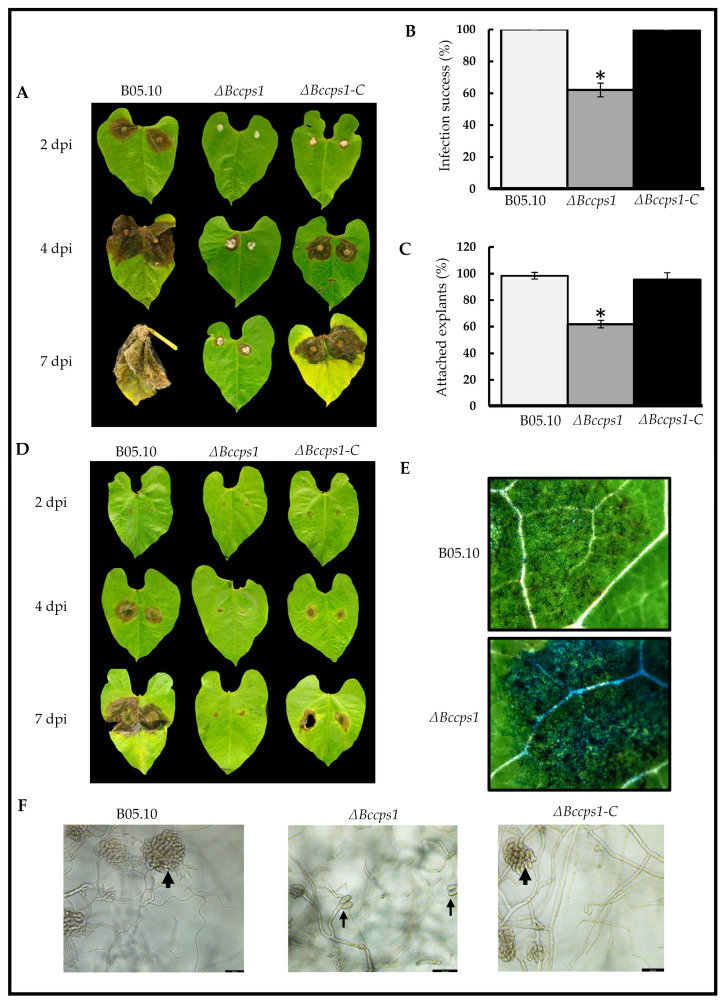
BcCps1 is involved in virulence. (**A**) Infection of bean leaves by mycelial explants of the parental (B05.10), mutant (Δ*Bccps1*) and complemented (Δ*Bccps1*-C) strains. Photos representative of the symptoms development at 2-, 4- and 7-days post inoculation (dpi) are shown. (**B**) Counting of the explants that succeeded at initiating infection. (**C**) Counting of the explants able to attach to the plant surface. Means and standard deviations were calculated from three independent experiments (*n* = 9). Stars indicate significant differences (*T*-test bilateral; *p* < 0.01). (**D**) With conidia. (**E**) Microscopy images of leaves collected from (**D**) at 2 dpi and colored with cotton blue to visualize fungal hyphae. (**F**) Development of infection cushions (large arrows) at 48 h of hyphal growth on plastic or glass surfaces in the parental and complemented strains. Development of hooks (thin arrows) in the mutant strain. Scale bars: 50 nm.

## Data Availability

The mass spectrometry proteomics data have been deposited to the ProteomeXchange Consortium via the PRIDE partner repository with the dataset identifier PXD033590.

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
