# Peer review of "Evidencing New Roles for the Glycosyl-Transferase Cps1 in the Phytopathogenic Fungus *Botrytis cinerea"

_jof, 2022, doi:10.3390/jof8090899_

Round 1

Reviewer 1 Report

This study reveals a role of glycosyl-transferase Cps1 in Botrytis cinerea. The function in hyphal expansion independent with conidia germination and early mycelial development was reported. A transgenic mutation line has been established to confirm the strongly impaired plant penetration ability in vivo and colonization as well as the formation of sclerotia in vitro.  The possible mechanism and cross-talk signaling are discussed well in the introduction and discussion sections. Some points can be improved.

1)     What are the black dots among pellets in Fig.2C? Scale bar is missing

2)     Concerning he structure in Fig.4A mutant, the size is not clear, what are these structure in the magnified circle, the numerous white clusters in the petri dish? However, the magnification rate is missing, if comparing to the left image it is hard to recognize. Again, scale bar is necessary.

3)     UV-irradiation was used in the experiment; however, no spectrum specifying can be found in any section of the manuscript.

Author Response

1. What are the black dots among pellets in Fig.2C? Scale bar is missing.

The black dots are the holes of the funnel holding the filter paper. Following filtration of the culture medium, they are visible through the wet paper. This is now explained in the figure legend.

2. Concerning the structure in Fig.4A mutant, the size is not clear, what are these structure in the magnified circle, the numerous white clusters in the petri dish? However, the magnification rate is missing, if comparing to the left image it is hard to recognize. Again, scale bar is necessary. 

Precisions have been added to the figure legend. The magnified images have been set to the same scale and a scale bar has been added.

3. UV-irradiation was used in the experiment; however, no spectrum specifying can be found in any section of the manuscript. 

The near UV-irradiation used is provided by a lamp shining at 365 nm. This precision has been included in the Material & Method section 2.2.

Reviewer 2 Report

Manuscript entitled “Evidencing new roles for the glycosyl-transferase Cps1 in the phytopathogenic fungus Botrytis cinerea” described by Blandenet et al. investigated the new roles of glycosyl-transferase Cps1 in Botrytis cinerea. The manuscript is well written, and the results had significance for expanding the new function of glycosyl-transferase. Minor revision need be addressed before it can be published.

1. Keywords: Gps1 replaced by glycosyl-transferase; Delete GT2.

Retain adhesion or virulence. Because the virulence to plants was equal to the adhesion ability to host plants.

2. Remove section 2.11 before 2.3. Firstly analyzed the informatics of the gene, then conducted gene deletion.

3. What about the resistance maker used for select complemented strains? G418?

4. Figure 1 moved to Supplementary files

5. Line 68-69. A. nidulans, F. verticilloides

6. Supplementary files should not be list as Non-published material.

7. Statistic analysis should be also conducted in Figure S5-S8.

Author Response

1. Keywords: Cps1 replaced by glycosyl-transferase; Delete GT2. Retain adhesion or virulence. Because the virulence to plants was equal to the adhesion ability to host plants.

GT2 has been deleted, Cps1 has been replaced by glycosyl-transferase and adhesion has been retained (virulence has been deleted).

2. Remove section 2.11 before 2.3. Firstly analyzed the informatics of the gene, then conducted gene deletion.

The section 2.11 is now section 2.3 and the following sections have all be renumbered accordingly.

3. What about the resistance maker used for select complemented strains? G418?

The resistance marker used is the neomycin resistance gene nptII. This was only indicated as « P7-neomycin plasmid » in former section 2.3, but it is now described in details in section 2.4.

4. Figure 1 moved to Supplementary files

This proposition is surprising to us because Fig.1 brings new phylogenetic information that deserves to be directly accessible to the reader. Furthermore, it contains structural and functional information about the BcCps1 protein that are very relevant to many people working in the fields of transmembranous GT2 enzymes and that are referred to in different parts of the manuscript, making the writing of these different parts simpler and clearer. We do wish to maintain Fig.1 where it is in the manuscript.

5. Line 68-69. A. nidulans, F. verticilloides

Line 68-69, Aspergillus nidulans has been replaced by A. nidulans and Fusarium verticilloides has been replaced by F. verticilloides.

6. Supplementary files should not be list as Non- published material.

By mistake, one figure not called in the manuscript was included as a supplementary figure. This figure has now been removed.

7. Statistic analysis should be also conducted in Figure S5-S8.

The statistics had been conducted for the data presented in these figures but were not shown in the first version of the manuscript. This has now been corrected.